# The role of community-level men's and women's inequitable gender norms on women's empowerment in India: A multilevel analysis using India's National Family Health Survey–5

Lakshmi Gopalakrishnan[1]*, Alison El Ayadi[2,3], Nadia Diamond-Smith[1,3]

1 Institute for Global Health Sciences, University of California, San Francisco, San Francisco, CA, United States of America, 2 Department of Obstetrics and Gynecology, University of California, San Francisco, San Francisco, CA, United States of America, 3 Department of Epidemiology and Biostatistics, San Francisco, CA, United States of America

* lakshmi.gopalakrishnan2@ucsf.edu

**Data Availability Statement:** Data is publicly available from DHS Website https://dhsprogram.

## Abstract

### Background

Lower empowerment of women is a critical social issue with adverse public health implications. In India, deeply ingrained gender norms shape a patriarchal structure that creates systemic disadvantages for women relative to men. These gender norms—socially constructed expectations about the roles, behaviors, and attributes of men and women—perpetuate inequality and limit women's opportunities.

### Objectives

The aim of this study was to examine the association between community-level men's and women's gender norms on women's empowerment in India. Women's empowerment was defined using four measures: freedom of movement, decision-making power, economic empowerment, and health empowerment.

### Methods

Using a nationally representative demographic health survey data from 2019–21 of 63,112 married women who participated in the women's empowerment module and 101,839 men surveyed, we constructed community-level men's and women's inequitable gender norms variables as our independent variable using attitudes towards wife-beating questions. We used random effects logistic regression models to examine if community-level men's and women's inequitable gender norms were independently associated with the different dimensions of women's empowerment.

com/Countries/Country-Main.cfm?ctry_id=57&c=
India&Country=India&cn=&r=4.

**Funding:** The author(s) received no specific
funding for this work.

**Competing interests:** The authors have declared
that no competing interests exist.

## Results

One standard deviation increase in community-level men's and women's inequitable gender norms was associated with reduced odds of freedom of movement, decision-making power, and health empowerment. No statistically significant association was observed between community-level men's and women's gender norms and economic empowerment.

## Conclusion

Inequitable gender norms are a risk factor that is negatively associated with several dimensions of women's empowerment. Our findings support our hypotheses that women's empowerment is impacted separately by men's and women's gender norms. Our study underscores the pressing need for concerted efforts to challenge and transform inequitable gender norms, paving the way for achieving gender equality and women's empowerment, as envisioned by the Sustainable Development Goals.

## Introduction

Gender norms are part of a larger construct of gender as a system, along with gender roles, gender socialization, and gendered power relations [1, 2]. Gender norms are "in the world"—they shape people's experience of their gender and their worldview. Depending on the culture, society, or reference group, gender norms are critical in constructing gender identities and lead to behaviors and actions that are considered appropriate for males and females, as well as what is considered masculine and feminine [1]. While the valuation of masculinity and femininity can vary depending on specific contexts or issues—for instance, femininity may be ascribed greater value in situations related to childbearing or upholding family honor—gender systems often tend to be patriarchal overall. In such systems, masculinity is generally ascribed greater value and power across most domains of social, economic, and political life [1]. This overarching patriarchal structure can persist even when femininity is valued in certain limited contexts. Gender norms can enable gender justice when they are egalitarian or can adversely impact gender justice when they are patriarchal [2].

Gender plays a crucial role in shaping various aspects of health and well-being across diverse societal contexts. India's patriarchal structure socially disadvantages women compared to men in India, reflected in their limited access to and control over resources, lower autonomy, restrictions in mobility, early marriage of girls, and gender-based violence [3]. Nearly a third of ever-married women have experienced some form of intimate partner violence (IPV), and about a quarter of women who faced IPV have experienced physical injuries related to IPV [4]. Culturally constructed patriarchal gender norms that endorse favoritism towards males and entrenched notions of power and patriarchy continue to perpetuate gender inequality in India [5]. The most apparent evidence of this is India's unevenly skewed sex ratios at birth (favoring boys), which was 907 girls per 1000 boys [6], and the persistent favoritism towards boys displayed in care-seeking practices, immunization, breastfeeding, and access to nutrition throughout childhood [7, 8]. These patterns continue in adulthood, where women often eat last in the household, do not get enough nutrition, and even have a higher risk of maternal morbidity and mortality due to family pressure to bear more children resulting from son preference [9–11]. Gender segregation marked by men and women eating separately and the practice of seclusion in the form of purdah/ghunghat (veil) and lack of mobility are

associated with early marriage, lower decision-making power for women, and lower access to economic resources [12].

Women's empowerment is a multidimensional construct defined as the "expansion of people's ability to make strategic life choices in a context where this ability was previously denied to them" [13]. It is codified within Sustainable Development Goal Five (SDG5)'s focus to empower women and girls to realize gender equality by 2030 [14]. The many dimensions of empowerment include economic, social, cultural, political, legal, and psychological empowerment [15]. Women's ability to achieve many of these outcomes is likely to be intricately linked to the society and norms of those decision-makers around them, making gender norms a key construct to explore in relation to empowerment.

Gender norms are important upstream determinants of gender inequality [16]. These norms could explain the low status of women and empowerment of women in Indian society, but this has not been empirically tested yet. The evidence on the association between community-level gender norms and women's empowerment is limited in the Indian context, potentially due to the paucity of comprehensive data on both men's and women's gender attitudes and norms, and how they interact to form community-level norms [17]. Few papers have empirically assessed this relationship in the Indian context: we found one dated paper that studied the effect of patriarchy on fertility from the 1990s [15] and two papers that studied the association between men's attitudinal norms and family planning use among married women [17, 18]. A recent qualitative study from Bangladeshi villages examined men's perspectives on gender equality and found men's shifting views on women's empowerment and masculinity were influenced by their self-interest, such as economic advancement and fear of repercussions, correlated with a decrease in intimate partner violence (IPV), reflecting changing gender norms. Further, few studies in African contexts have shown that men's attitudes about gender equality are associated with condom usage to prevent HIV [19, 20]. Still, limited studies have examined the contextual influence of men's and women's inequitable gender norms on women's empowerment in South Asia.

In this study, we examined the associations between community-level men's and women's inequitable gender norms on the various dimensions of women's empowerment measured at the individual level. We posited that community-level gender norms are a byproduct of men's beliefs about women's roles and duties––especially in the context of cultural values, traditional family roles, gender values, and gender order. Since women also live in the same communities as men, we hypothesized that women's attitudes towards gender equality might also add to the normative environment, which could influence women's empowerment. Specifically, we hypothesized that women residing in communities with higher inequitable men's gender norms and inequitable women's gender norms (modeled independently) will be more likely to have lower freedom of movement, lesser decision-making power, lower economic resources, and lower health empowerment.

## Theoretical framework

We used Kabeer's framework extended by Yount as a multi-level approach to studying women's empowerment (Fig 1) [13, 21]. In this framework, women's empowerment is conceptualized as related to economic resources (labor force participation, having a bank account, savings), human resources (education), agency (decision-making, freedom of movement), and achievements (health and nutrition outcomes) [13]. Factors at the individual and community-level distinctively influence women's empowerment measures [21]. Empowerment at the individual level is an outcome of intersections of community, household, and individual level factors. The community-level factors (including gender norms) often influence the attitudes of both women and men towards various aspects of women's empowerment.

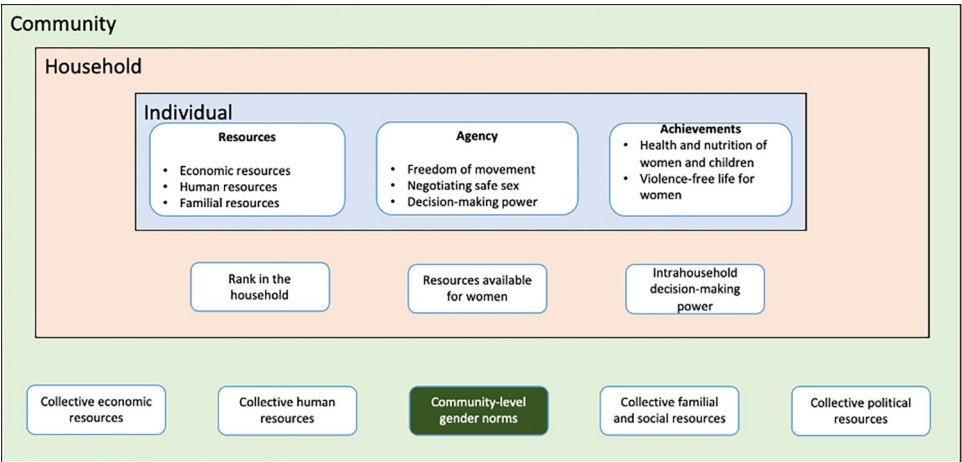

**Fig 1. Multi-level framework to study women's empowerment (adapted from Kabeer and Yount)** [13, 21].

## Methods

### Data source

We used the most recent round of India's National Family Health Survey (NFHS-5) 2019–21 data. The NFHS-5 data was collected from a nationally representative, stratified random two-stage sampling between June 2019 and April 2021 across 707 districts in 28 States and eight Union Territories. Census 2011 was used as a sampling frame for the selection of primary sampling units (PSU)––villages in rural areas and Census Enumeration Blocks (CEBs) in urban areas. More detailed sampling information is provided in the NFHS-5 report [4]. The NFHS is a multi-topic survey that covers different modules related to demographics, economic activity, household work, migration, mass media and internet exposure, awareness of pregnancy, HIV, family planning methods, health-seeking behaviors, and child health and nutrition. For a sub-sample of households, the NFHS also covers several domains of empowerment measures, including decision-making, freedom of movement, economic empowerment, and health empowerment.

### Analytical data

The NFHS typically selects primary sampling units (PSUs) that align with villages in rural settings and census enumeration blocks in urban regions. These PSUs consist of clusters of households sharing similar geographical, ecological, and cultural characteristics, referred to as communities in this article. The study aggregated variables measured at the community level to the PSU level. The NFHS-5 is part of the Demographic and Health Surveys (DHS) Program and contains three core questionnaires: a household questionnaire, a men's questionnaire, and a women's questionnaire. For this study, we used information from the women's and men's questionnaires described below.

The women's data is restricted to currently married women (n = 512,408) because of our interest in women's empowerment indicators, such as joint decision-making and co-ownership of economic resources with the husband. Only one woman was selected from each sampled household to answer the women's empowerment and attitudes towards wife-beating module, which reduced our sample to 76,910 currently married women respondents aged 15–49. After conducting a complete case analysis, our final analytic sample for this study was 63,112 married women from 8929 PSUs, which we defined as our analytical sample. We used

this final dataset to construct community- and individual-level indicators of women's empowerment as well as individual covariates. Each PSU had on average 7 women, ranging from a minimum of 1 through 20 women.

The men's data includes a sample of 101,839 respondents aged 15–54 across 9102 PSUs. Each PSU had an average of 4 men, ranging from a minimum of 1 through 15 men, that we used to construct community-level exposures reflecting gender inequitable norms related to attitudes towards wife beating.

In the survey, the data of men were not collected from all households, so the variable was aggregated at the community level. Community-level inequitable gender norms were created and aggregated using men's dataset and then merged with women's dataset using PSU-level identifiers. Similarly, the data on the attitudes towards wife beating among women were calculated from women's data but aggregated at the community level to get a variable from women comparable with men. All estimates in this study are based on the weighted sample, and numbers are unweighted. Since we used de-identified data publicly available from the DHS website [22], we were exempt from seeking ethical approvals. We did not have access to identifying individual participant data.

## Measures

**Dependent variables.** The dependent variables for the analysis use four dimensions of women's empowerment, constructed at the individual (woman) level. Our operationalization of women's empowerment dimensions draws on established frameworks and measures in the literature. The freedom of movement and decision-making power measures are adapted from the Demographic and Health Surveys methodology [23], similar to those used in papers published previously [24, 25]. The economic empowerment index combines indicators suggested by Golla et al. [26] and Kabeer [13]. Our health empowerment index incorporates measures of reproductive health and HIV knowledge used previously [25, 27]. The choice of these four dimensions is grounded in Kabeer's conceptualization of empowerment as resources, agency, and achievements, as also our theoretical framework [13].

Below, we explain how each of the variables was operationalized for the analysis and dichotomized as follows:

1. *Freedom of movement*: We considered women to be empowered if they had the freedom of movement if they were allowed to go alone to all three places—market, health facility, and outside the village (coded as 1) and 0 otherwise.

2. *Decision-making power*: We dichotomized the variable and considered women as empowered (coded as 1) if they participated in all household decisions either alone or jointly with their husband regarding the husband's earnings, woman's healthcare, major household purchases, and visits to the woman's family and 0 otherwise.

3. *Economic empowerment*: First, we dichotomized each of the following six questions–– whether the woman had worked in the past 12 months (yes = 1; no = 0), whether she had her own money she could decide to use (yes = 1; no = 0), bank account with savings she can use (yes = 1; no = 0), mobile phone she can use (yes = 1; no = 0), whether the woman owned the house either alone or jointly coded as 1, and 0 otherwise, and finally whether she owned land either alone or jointly coded as 1, and 0 otherwise. Responses across all these questions were summed to create a score ranging from 0–6. We classified women with at least 50% access to economic resources being economically empowered (coded as 1) and 0 otherwise.

4. *Health empowerment index* was constructed as a summative index and then dichotomized with the median score as a cut-off. First, each of the knowledge questions on fertility, HIV, and modern family planning methods were recoded as follows: whether a woman knew the most fertile period was between the two menstrual period cycles (yes = 1; 0 otherwise), each of the five questions on HIV was coded as 1 if they answered correctly and 0 otherwise. Similarly, we recoded correct modern family planning methods into a dichotomous variable if they answered correctly (1 = correct; 0 otherwise). All these items were summed and dichotomized: women with greater than median score of correct answers were defined as empowered (coded as 1) and 0 otherwise.

**Independent variables.**  As mentioned previously, the primary sampling unit (PSU) of the NFHS generally coincides with villages in rural areas and census enumeration blocks in urban areas. Since the PSUs are a cluster of households with a common geographical, ecological, and cultural environment, we defined this as community-level in our exposure variable. Community-level inequitable gender norms were constructed from men's data and combined with women's data using PSU-level identifiers. Likewise, women's attitudes toward wife beating were computed from women's data and aggregated at the community level for comparison with men's data. The main independent variables of interest in our study were community-level inequitable gender norms, defined separately using men's and women's "collective attitudinal norms" [28–30] and described below:

1. *Community-level inequitable men's gender norms*. Since there is no gold standard measure for gender norms, we relied on proxy measures of gender norms [16]. Community-level inequitable men's gender norms variables were constructed from seven survey questions posed to men separately that measure the respondents' views regarding the acceptability of attitudes toward wife beating. Survey items included if they agreed that a husband was justified in inflicting violence towards his wife under each of the following seven circumstances: she goes out without telling him, she neglects the house or the children, she argues with him, she refuses to have sex with him, she doesn't cook food properly, he suspects her of being unfaithful, and she shows disrespect for her in-laws. For each item, respondents indicated whether they agreed, disagreed, or "don't know." Items were coded such that a response justifying violence as acceptable inequitable was coded as 1 and 0 otherwise. "*Don't know*" responses were combined such that the absence of an affirmative response indicated inequitable gender attitudes to ensure that we erred on the side of being more conservative. Items were added to create a summative scale from 0 to 7, with higher scores representing more inequitable gender attitudes (Cronbach's alpha 0.84). The men's data was aggregated as a mean at the community or PSU-level. To ease the interpretation of the regression, we standardized men's attitudinal gender norms as community-level men's inequitable gender norms by rescaling the variables to a mean of zero and a standard deviation of one.

2. *Community-level inequitable women's gender norms*. Community-level inequitable women's gender norms variables were constructed from seven survey questions posed to women that measured the respondents' views regarding the acceptability of attitudes toward wife beating. Items were added to create a summative scale from 0 to 7 like that in the men's data. Since the outcome was also measured among women, we created non-self-means by removing the index woman when averaging the responses and collapsed at the PSU level to create a variable from women. Cronbach's alpha for women's gender norms scale was 0.85. We also standardized women's inequitable gender norms to ease interpretation.

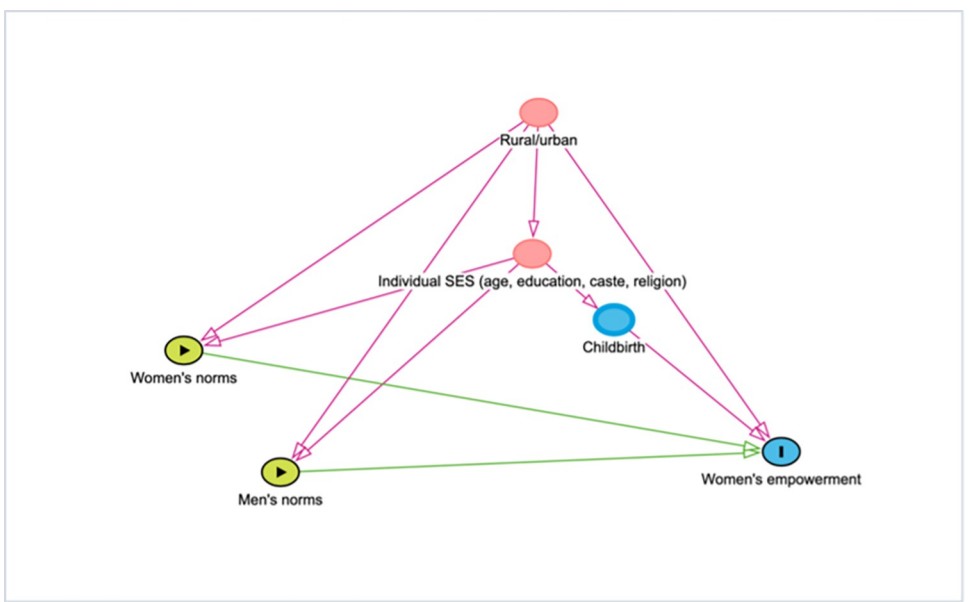

**Fig 2. Directed Acyclic Graph to study the association of community-level men's and women's gender norms and women's empowerment.**

**Covariates.** *In* our statistical analysis, we considered several potential individual-level factors that could influence women's empowerment and men's attitudes. These factors, include age, women's education, wealth status, religion, caste, place of residence, and children ever born based on previous studies on women's empowerment [18, 24, 31]. We used a Directed Acyclic Graph (DAG) (Fig 2) to visually represent the hypothesized causal relationships between variables. This approach, based on causal inference theory [32] (Pearl, 2009), helped us identify potential confounders and avoid overcontrol bias. We specified women's age as a continuous variable to capture life-course effects on women's empowerment. Women's education (in years of schooling) was also specified as a continuous variable given its extensive association with women's empowerment. We constructed the wealth index, a composite index reflecting a household's living standard and assets using principal components analysis [33]. Wealth scores were generated, divided into 5 quintiles, from poorest (1) to wealthiest (5). Religion was specified as a categorical variable––Hindus (as reference category), Muslims, Christians, Sikhs, and others. Similarly, caste was specified as a categorical variable––General caste (reference category), Scheduled Caste/Scheduled Tribe, Other Backward Classes. The place of residence was coded as urban or rural. Child ever born was specified as a binary variable and was included in the multivariable regression model to capture the potential influence of childbearing on women's status within the household.

## Analysis

We first computed sample descriptive characteristics to examine each covariates' distribution. We generated proportions of outcome variables, including freedom of movement, decision-making, economic empowerment, and health empowerment. We also described the weighted proportion of men's and women's responses to attitudes towards wife beating and the distribution of the constructed community-level men's and women's gender norms variables and the correlation between them.

For the multivariable analysis, we used a three-level logistic random effects model to examine the association of patriarchal gender norms at the community level and each of the measures of women's empowerment at the individual level, controlling for covariates. Multi-level models allow us to simultaneously run regression models for each data level, considering the lack of independence of the nested observations and residuals. Multi-level models partition the variance in the outcome variable at the individual versus cluster levels. The statistical significance of covariance was estimated using the likelihood ratio test. All significance tests were two-tailed, and statistical significance was defined at the 5% alpha level. All data were analyzed using Stata Version 15.1 [34].

The model specification is as follows:

$$y_{ijk} = \beta_0 + \beta_{1,jk}\boldsymbol{M_{jk}} + \beta_{2,jk}\boldsymbol{F_{jk}} + \beta_{3,jk}\boldsymbol{X_{ijk}} + v_k + \mu_{jk}$$

Where $y_{ijk}$ represents the individual level women's empowerment scores (across each domain separately) of a woman $i$ in community (defined by PSU) $j$, in state $k$. $\boldsymbol{M_{jk}}$ and $\boldsymbol{F_{jk}}$ represents male and female inequitable gender norms variables measured at the level of the community $j$ in each state k from men and women, respectively. $\boldsymbol{X_{ijk}}$ represents a vector of individual-level exposures and covariates for each woman at the individual nested within a community $j$, in state $k$. $v_k$ and $\mu_{jk}$ are random coefficients representing the residual variation at the community and state level, respectively.

Four multilevel logistic models were estimated, one for each dependent variables: freedom of movement, decision-making power, economic empowerment, and health empowerment. In each model, the first level is the individual, and the second level is the community, and the third level is the state. State was added to account for additional clustering associated with the study's design.

## Results

### Descriptive analysis

In Table 1, we present the socio-demographic characteristics of women in our analytical sample. The mean age of women in our sample was 33.5 years (SD:8.3). A quarter of the women had no education (24.3%), 13% had completed primary school education, and nearly 15% had completed higher education. The majority belonged to the Hindu religion (83.8%). Regarding respondent's caste composition, most women (47.0%) belonged to the Scheduled Caste/Scheduled Tribe group, considered one of the most marginalized groups, followed by Other Backward Class (31.1%). One-third of women (33.1%) belonged to urban areas. Most of the women (36.9%) had two children; less than 10% of women had not had any children yet.

### Men and women's responses to attitudes towards wife beating scale

Table 2 presents a weighted proportion of men's and currently married women's responses regarding attitudes towards wife beating. Among both men and women, the most prevalent belief for which wife beating was justified was if women showed disrespect to their in-laws (32.7% men and 33.2% women).

The least prevalent norm was again common for both men and women––nearly 10.0% of men and women endorsed wife beating if wife refuses sex with their husband.

Table 3 displays the mean and standard deviation of the community-level male and female inequitable gender norms; the mean of community-level men's inequitable norms was 1.2 (SD ±1.3), and the mean of women's gender norms was 0.7 (SD ±0.6). The correlation between both norms was 0.3.

**Table 1. Socio-demographic characteristics of married women (15–49 years) in the analytical sample, National Family Health Survey-5, 2019–21 (n = 63,112).**

| Variable | Weighted % /mean (SD) | n |
|---|---|---|
| Age in years | 33.5(8.4) | 63,112 |
| Women's education | | |
| No education | 24.3 | 16,047 |
| Primary school education | 29.5 | 19,089 |
| Secondary school education | 31.4 | 19,860 |
| Higher education | 14.8 | 8,116 |
| Religion | | |
| Hindu | 83.8 | 49,504 |
| Muslim | 10.9 | 5,973 |
| Christian | 2.4 | 4,571 |
| Sikh | 1.7 | 1,440 |
| Other | 1.1 | 1,624 |
| Caste | | |
| General Caste | 21.9 | 12,639 |
| Other Backward Class | 31.1 | 24,229 |
| Scheduled Caste/Scheduled Tribe | 47.0 | 26,244 |
| Wealth quintile | | |
| Poorest | 15.5 | 11,474 |
| Poor | 18.9 | 13,481 |
| Middle | 21.2 | 13,531 |
| Rich | 21.9 | 12,830 |
| Richest | 22.3 | 11,796 |
| Place of residence | | |
| Urban | 33.1 | 15,968 |
| Rural | 66.9 | 47,144 |
| Number of children born | | |
| None | 9.0 | 5,582 |
| 1 child | 19.3 | 11,892 |
| 2 children | 36.9 | 22,231 |
| 3 children | 19.2 | 12,687 |
| ≥4 children | 15.4 | 10,720 |

**Table 2. Weighted proportion of men's and women's responses to attitudes towards wife beating from men and currently married women in National Family Health Survey-5, 2019–21.**

| Justify wife-beating for the following reasons | Individual men's responses | | Individual women's responses | |
|---|---|---|---|---|
| | Weighted % | n = 101,839 | Weighted % | n = 76,910 |
| Goes out without telling husband | 16.1 | 16,428 | 20.2 | 14,283 |
| Neglects the house or children | 23.1 | 23,503 | 28.7 | 19,734 |
| Argues with husband | 21.3 | 21,685 | 23.4 | 16,297 |
| Refuses to have sex with husband | 11.5 | 11,697 | 11.9 | 8,589 |
| Does not cook food properly | 11.2 | 11,362 | 14.5 | 10,527 |
| Suspects wife of being unfaithful | 24.7 | 25,119 | 21.2 | 14,814 |
| Shows disrespect to in-laws | 32.7 | 33,288 | 33.2 | 23,601 |

**Table 3. Distribution of constructed community-level men's and women's gender norms variables in the analytical sample.**

| Community-level mean gender norms | n | Mean (SD) | Range |
|---|---|---|---|
| Community-level men's inequitable gender norms | 63112 | 1.2 (1.3) | 0–7 |
| Community-level women's inequitable gender norms | 63110 | 0.7 (0.6) | 0–4 |
| Correlation between community-level men's gender norms and women's gender norms | 0.3065 | | |

\* Note: n = 63110 for women because 2 PSUs have only 1 woman each and due to non-self-mean calculation, those PSUs/communities are dropped from the sample

Table 4 shows the weighted proportion of dependent variable measures among the analytical sample of married women. About 44.6% of women were allowed to go alone to all three places––market, health facility, and places outside the community. Nearly two-thirds of women (66.4%) participated in decision-making alone or jointly with their husbands. About 38.6% of women had at least equal to or more than median-level knowledge of questions on HIV, fertility, and family planning. About 65% women had at least the median-level or greater economic empowerment score.

## Multilevel multivariable analysis

Table 5 presents our multilevel random-effects logistic regression models for the four different empowerment measures. As hypothesized, men's inequitable norms were negatively associated with almost all measures of women's empowerment. We found that one standard deviation increase in community-level men's gender norms was associated with reduced odds of freedom of movement (AOR:0.91;p<0.001), decision-making power (AOR: 0.89;p<0.001), and economic empowerment(AOR: 0.96;p<0.05), after controlling for individual covariates. No statistically significant association was found between community-level prevalence and economic empowerment.

Further, we had a similar finding in relation to women's inequitable gender norms. One standard deviation increase in community-level women's gender norms was associated with reduced odds of freedom of movement (AOR: 0.95;p<0.001), decision-making power (AOR: 0.89; p<0.001), and health empowerment (AOR: 0.97; p<0.001) controlling for individual covariates. We did not find a statistically significant association between women's inequitable gender norms and economic empowerment.

## Discussion

Our study examined the association between community-level inequitable gender norms, as expressed by both men and women separately, and four dimensions of women's

**Table 4. Descriptive statistics of dependent variables (women's empowerment) chosen as dependent variables (n = 63,112).**

| Empowerment measure | Description | Weighted % | |
|---|---|---|---|
| Freedom of Movement | Women usually allowed to go alone all the three places (market, outside the village, friends/relatives inside the village) | 44.6 | 28,146 |
| Decision-making Power | Women make all the decisions either alone/jointly with husband | 66.4 | 42,684 |
| Health Empowerment | Women who have at least the median-level knowledge of correct answers to the health-related questions on HIV, fertility, and family planning | 38.6 | 23,764 |
| Economic Empowerment | Women who have at least the median-level economic empowerment scores | 65.6 | 19,356 |

**Table 5. Multivariable mixed effects exploring the association between community-level male and female inequitable gender norms on different dimensions of women's empowerment (odds ratios, 95% confidence interval).**

| | Model 1 | Model 2 | Model 3 | Model 4 |
|---|---|---|---|---|
| | Freedom of movement alone | Decision-making power | Economic empowerment | Health empowerment |
| **Community-level men's gender norms (1 SD)** | 0.91*** | 0.89*** | 1.01 | 0.96* |
| | (0.88–0.94) | (0.86–0.92) | (0.98–1.05) | (0.93–0.99) |
| **Community-level women's gender norms (1 SD)** | 0.95*** | 0.89*** | 0.99 | 0.97* |
| | (0.92–0.97) | (0.87–0.91) | (0.96–1.01) | (0.95–1.00) |
| **Age (in years)** | 1.05*** | 1.03*** | 1.03*** | 1.008*** |
| | (1.04–1.05) | (1.02–1.03) | (1.03–1.04) | (1.00–1.011) |
| **Women's education (in years)** | 1.03*** | 1.02*** | 1.07*** | 1.08*** |
| | (1.02–1.03) | (1.02–1.03) | (1.06–1.07) | (1.08–1.09) |
| **Rural (Ref: Urban)** | 0.70*** | 0.91* | 1.17*** | 0.92* |
| | (0.65–0.75) | (0.84–0.98) | (1.08–1.25) | (0.86–0.99) |
| **Caste (Ref: General caste)** | | | | |
| Scheduled Caste/Scheduled Tribe | 1.06 | 1.04 | 1.17*** | 0.97 |
| | (0.99–1.12) | (0.97–1.11) | (1.09–1.24) | (0.91–1.04) |
| Other Backward Class | 0.93* | 0.99 | 1.02 | 0.94 |
| | (0.88–0.99) | (0.93–1.05) | (0.95–1.08) | (0.89–1.00) |
| **Religion (Ref: Hindus)** | | | | |
| Muslims | 0.65*** | 0.86*** | 0.77*** | 0.86*** |
| | (0.60–0.71) | (0.79–0.93) | (0.70–0.83) | (0.78–0.94) |
| Christian | 1.08 | 1.25*** | 1.03 | 0.96 |
| | (0.95–1.23) | (1.09–1.43) | (0.90–1.18) | (0.84–1.09) |
| Sikh | 0.94 | 1.06 | 1.17 | 1.21 |
| | (0.77–1.15) | (0.861–1.317) | (0.953–1.450) | (0.997–1.477) |
| Other | 1.12 | 1.10 | 1.02 | 0.87 |
| | (0.96–1.309) | (0.94–1.30) | (0.87–1.21) | (0.75–1.03) |
| **Wealth index (Ref: Poorest quintile)** | | | | |
| Poorer | 0.95 | 0.94 | 1.10** | 1.06 |
| | (0.89–1.02) | (0.88–1.00) | (1.03–1.17) | (0.99–1.14) |
| Middle | 0.85*** | 0.95 | 1.21*** | 1.19*** |
| | (0.79–0.92) | (0.88–1.03) | (1.13–1.30) | (1.11–1.28) |
| Rich | 0.83*** | 0.87*** | 1.20*** | 1.26*** |
| | (0.77–0.90) | (0.80–0.94) | (1.109–1.299) | (1.16–1.36) |
| Richest | 0.85*** | 0.88* | 1.55*** | 1.52*** |
| | (0.78–0.94) | (0.81–0.98) | (1.41–1.71) | (1.39–1.67) |
| **Child ever born** | 1.07*** | 1.04*** | 1.02* | 0.99 |
| | (1.05–1.09) | (1.02–1.06) | (1.00–1.04) | (0.98–1.01) |
| **ICC at State level** | 0.18 | 0.04 | 0.09 | 0.07 |
| **ICC at PSU-level** | 0.37 | 0.26 | 0.28 | 0.28 |
| **Observations** | 63,110 | 63,110 | 63,110 | 63,110 |
| **Number of states** | 36 | 36 | 36 | 36 |
| **Number of PSUs** | 8,929 | 8,929 | 8,929 | 8,929 |
| 95%CI in parentheses | | | | |
| *** p<0.001, ** p<0.01, * p<0.05 | | | | |

empowerment using nationally representative data from NFHS-5: freedom of movement, decision-making power, health empowerment, and economic empowerment. We found significant negative associations between inequitable gender norms at the community-level and three dimensions of empowerment (freedom of movement, decision-making power, and health empowerment), while the association with economic empowerment was not significant. Interestingly, effect sizes were similar for both men's and women's inequitable norms, suggesting that the overall normative environment, rather than the gender of those holding the beliefs, is crucial in influencing women's empowerment.

Freedom of movement: Our analysis revealed a significant negative association between community-level inequitable gender norms and women's freedom of movement. Specifically, for one standard deviation increase in community-level inequitable men's and women's gender norms, the odds of women having freedom of movement decreased by approximately 8% and 5%, respectively. This finding aligns with previous research by Jayachandran [5] and Marcus [35], highlighting how cultural norms restrict women's freedom of movement and access to education, healthcare, and employment opportunities. This mechanism likely involves societal expectations about safeguarding women's "purity", reputational risks, and fears about women's safety outside the home. In communities with more inequitable norms, there may be greater social sanctions against women traveling alone, reinforced by both men and women in the community. This internalization of restrictive norms by women themselves highlights the pervasive nature of these cultural expectations.

Decision-making power: Each standard deviation increase in community-level men's and women's inequitable norms was associated with an 11% decreased odds of women's participation in household decisions. This finding is consistent with Duflo's review [36], which highlighted that traditional gender norms often limit women's participation in household decision-making, including decisions about children's education and healthcare. Our findings on the negative association between inequitable gender norms and women's decision-making power align with broader regional patterns in South Asia, where patriarchal norms significantly constrain women's agency in crucial life decisions [37]. The internalization of patriarchal norms by both men and women, leading to the expectation that men should be the primary decision-makers in the household. These norms are often deeply entrenched and manifest in various ways. As highlighted in regional studies, they seek to confine women to narrow roles within the domestic sphere and stigmatize those who breach these expectations [37]. The impact extends beyond the household, influencing women's ability to make decisions about employment, migration, and other aspects of their lives. The persistence of these norms may be reinforced through social interactions, family structures, and community expectations, creating a cycle where women's voices are systematically undervalued in household matters.

Health empowerment: Each standard deviation increase in men's and women's inequitable norms was associated with a 4% and 3% decrease in the odds of women having high health empowerment, respectively. This aligns with findings from Okigbo et al. in Nigeria [38]O, where improvements in gender-equitable attitudes were associated with increased adoption and sustained use of modern contraceptives. The mechanism underlying this association may involve restricted access to health information and services in communities with more inequitable gender norms. Women in these communities may face barriers in discussing health issues openly, seeking healthcare independently, or making autonomous decisions about their own health and that of their children. Additionally, in contexts with highly inequitable norms, there may be less emphasis on women's health education, further limiting their health empowerment.

Economic empowerment: Interestingly, we did not find a significant association between community-level inequitable gender norms and women's economic empowerment. This result

diverges from some previous research, including the Growth and Economic Opportunities (GrOW) programme, which funded 14 research projects across more than 50 low-income countries between 2013 and 2018, providing valuable insights into this complex relationship [35]. GrOW-supported research consistently identified gender discriminatory social norms as a significant barrier to women's economic empowerment. Norms about mobility and respectability significantly constrain women's economic activities [35].

Our contrasting finding might be explained by the complex nature of economic empowerment in the Indian context. Even when women have access to economic resources, deeply entrenched gender norms—shaped by religious beliefs, tribal governance structures, and historical community practices—can significantly constrain women's economic roles and opportunities [39]. Taken together, these findings suggest that economic empowerment may operate somewhat independently of other forms of empowerment, highlighting the multidimensional nature of women's empowerment and the need for nuanced approaches in its study. This complexity calls for further research to understand the specific mechanisms through which gender norms interact with economic factors in different cultural contexts, potentially explaining the lack of significant association in our study.

The similar effect sizes for men's and women's norms suggest that the overall normative environment, rather than the gender of those holding the beliefs, is the critical factor in influencing women's empowerment. This underscores the significant influence of normative environment on women's empowerment, demonstrating that the impact persists regardless of whether men or women are the primary upholders of patriarchal norms and regressive attitudes in a given community. This suggests that women, like men, can internalize and perpetuate patriarchal norms, contributing equally to the maintenance of gender inequalities. Further, similar impact of men's and women's norms also likely suggest that harmful gender attitudes are transmitted and reinforced through various social interactions and institutions, not just through male-dominated structures. Overall, our findings point to the need for comprehensive family-level and couples-level interventions aimed at addressing harmful gender norms to promote the empowerment of women and prevent gender-based inequities effectively. These interventions should focus on reshaping the overall normative environment rather than targeting only one gender group.

Overall, our findings align consistently with research from other low-and-middle-income countries, highlighting the adverse impacts of inequitable gender norms and offering significant contributions to the existing literature. Buffarini and colleagues established the importance of gender norms and socio-economic stratification by illustrating how the effects of restrictive gender norms hurt low-income girls the most on a range of health behaviors and outcomes, including smoking, weight, violence, happiness, and mental health in Brazil [40]. In a multi-level longitudinal study in Nigeria, Okigbo et al. found that improvement in gender norms, as evidenced by changes in gender-equitable attitudes towards household decision-making, and couples' family planning decisions, and family planning self-efficacy at the individual and neighborhood levels were associated with increased adoption and sustained use of modern contraceptive [38]. Inequitable gender norms significantly impact women's access to and use of financial services across developing countries [41]. A review by Duflo highlighted that traditional gender norms often limit women's participation in household decision-making, including decisions about children's education and healthcare [36]. Another review found that cultural norms restricting women's mobility significantly impact their ability to access education, healthcare, and employment opportunities [5]. Harmful gender norms have been widely recognized as significant barriers to women's empowerment across various cultures and societies.

Out study contributes to the limited body of research examining the association between community-level factors, particularly men's attitudes/norms, and their correlation with women's empowerment. The findings presented in our study are broadly consistent with previous studies on the effect of norms and different measures of women's empowerment [42–44], except for our results on norms and economic empowerment. In contexts where patriarchal systems are strong, such as India, analysis of community norms and their effect on various dimensions of women's empowerment provides important evidence of the need to address women's empowerment not only at the individual level but also at the community level.

To further advance this field of study, researchers and practitioners can employ various quantitative and qualitative approaches to analyze dimensions of women's empowerment at the community-level. Quantitative approaches could include––a. Household surveys: Implement surveys that capture both individual and community-level variables. These should include measures of perceived community norms alongside individual attitudes and behaviors. b. Social Network Analysis: Map women's formal and informal networks within the community and analyze the structure and strength of these networks in relation to empowerment outcomes. c. Time-use studies: Conduct community-wide time-use surveys to understand gendered patterns of work, leisure, and decision-making. Qualitative methods could include––a. Focus Group Discussions (FGDs): Use vignettes or scenarios to elicit community perspectives on women's roles. Conduct separate FGDs for men and women, as well as mixed-gender groups, to understand gender norms and observe power dynamics. b. Key Informant Interviews: Engage with community leaders, service providers, and other influential figures to understand institutional factors affecting women's empowerment. c. Participatory Rural Appraisal (PRA) techniques: Employ community mapping exercises to identify gendered spaces and resources and use seasonal calendars to understand the gendered division of labor. By employing a combination of these methods and adapting it their specific cultural context, researchers can gain a comprehensive understanding of the community-level factors influencing women's empowerment.

## Policy implications

Our findings have important implications for including both men and women in transforming inequitable gender attitudes and norms. This is particularly salient to focus early on given that recent research from north India suggests that gender attitudes, defined as the appropriate roles and rights of women and girls, form at an early age and are influenced by their parents' discriminatory gender attitudes––highlighting the role of intergenerational transmission in the formation of gender attitudes [45]. Researchers found that when a parent held a more discriminatory gender attitude, their child was about 11 percentage points more likely to have that same view [45]. The intergenerational transmission of gender attitudes is a well-documented phenomenon across various cultural contexts. While our initial reference focused on research from north India [45], numerous studies globally have corroborated this finding: A study using data from the British Household Panel Survey of children aged 11–15 found that children's gender role attitudes were significantly influenced by their parents' attitudes, with this effect persisting into adulthood [46]. Research in 37 countries using World Values Survey data showed that individuals' gender ideology is significantly associated with their parents' division of household labor during childhood, demonstrating the cross-cultural nature of this transmission [47]. These studies collectively underscore the pervasive nature of intergenerational transmission of gender attitudes across diverse cultural settings. They highlight the importance of targeting interventions towards both men and women, starting at an early age, to break the cycle of inequitable gender norms and foster more egalitarian gender attitudes.

Without intervening early to change gender norms and attitudes, it may be challenging to achieve the Sustainable Development Goal of achieving gender equality and women's empowerment.

Furthermore, these findings highlight the importance of engaging men to improve women's status and reduce gender inequality, ultimately improving women's and children's health and lives more broadly. There is a growing body of evidence on engaging men in improving a range of women's health outcomes, including gender-based violence [48–50]. It may be also important to consider interventions that engage not only male partners of women but also other family and community members, such as in-laws, brothers, community leaders, among other stakeholders. Many interventions aiming to support women's empowerment primarily target women or minimally engage other household member—however, a more specific focus on changing community-level gender norms is needed. One study in India addressed social and gender norms to reduce women's anemia rates and found that the intervention increased the odds of women having diverse diets and uptake of iron-folic acid supplements [51–54]; similar approaches could be used to address women's empowerment itself, which would likely have downstream effects on many health outcomes, as well as benefits for the women themselves and society more broadly.

## Strengths and limitations

Our study has some limitations. The causal relationship between the normative environment and women's empowerment measures is difficult to establish. As with all cross-sectional studies, we cannot know the direction of the association between women's empowerment and community-level norms. However, community level norms are likely to not change too quickly and, thus, are likely to have existed before the individual woman experienced the outcomes of interest. Gender norms also facilitate the formation of gender-role attitudes, which are beliefs held in a society that defines what is "appropriate behavior for men and women," and we are using an aggregate measure of men's and women's attitudes as a proxy for inequitable gender norms. Aggregating attitudes is not the same as aggregating what others in the community approve or disapprove of and what others in the community do, but it could still be a useful proxy when norms data are limited [16].

Gender norms have been gaining much attention in public health lately, without the commensurate development of measures needed to understand the complex role of gender norms in shaping behavior. To our knowledge, there is no consensus-based scale to measure gender norms, and no standard exists for quantitatively measuring beliefs and attitudes across cultures and over time [55]. Quantitative data and measures needed to estimate the influence of gender norms on the empowerment of women and health outcomes are in a nascent stage. Hence, the results of this study need to be interpreted cautiously. Another limitation is that we rely on self-reported responses from survey questions to assess attitudes that might be subject to social desirability bias. Finally, consolidating women's empowerment into a single-scale measure presents several limitations, such as potential measurement errors and the risk of overlooking the multifaceted nature of the concept [56].

Despite limitations, we leveraged an existing national-level dataset to consider how community-level norms influence individual outcomes, which has provided strong evidence of the importance of gender norms for women's empowerment, adding to the limited evidence to date on this topic. Further, by examining men's and women's inequitable gender norms separately, we were able to analyze the associations between gender-specific community-level norms and various dimensions of women's empowerment. This approach allowed us to observe differences in how men's and women's gender attitudes at the community level relate

to women's empowerment outcomes, though it does not provide insight into the mechanisms by which these attitudes are formed or internalized. Instead of using one composite index to measure empowerment, we examined them separately to understand better how norms influence these different dimensions of women's empowerment.

## Conclusion

Our study suggests that community-level inequitable gender norms, held by both men and women, are significantly associated with lower odds of freedom of movement, household decision-making power, and health empowerment. This research contributes significantly to understanding the complex relationship between gender norms and women's empowerment, particularly within the context of low- and middle-income countries like India. The lack of association between norms and economic empowerment highlights the multifaceted nature of empowerment and the need for nuanced, context-specific approaches in its study. Future research should explore the specific pathways explaining these associations and investigate whether similar relationships exist in other settings and for different outcomes. Employing a combination of quantitative and qualitative approaches can provide a more comprehensive understanding of the community-level factors influencing women's empowerment. Our research emphasizes the urgent need for comprehensive interventions targeting harmful gender norms at both family and community levels. These interventions should focus on reshaping the overall normative environment rather than targeting only one gender group It is imperative to involve both men and women in these efforts, given their shared responsibility in perpetuating or challenging prevailing gender norms.

## Author Contributions

**Conceptualization:** Lakshmi Gopalakrishnan.

**Formal analysis:** Lakshmi Gopalakrishnan.

**Methodology:** Lakshmi Gopalakrishnan.

**Supervision:** Alison El Ayadi, Nadia Diamond-Smith.

**Writing – original draft:** Lakshmi Gopalakrishnan.

**Writing – review & editing:** Lakshmi Gopalakrishnan, Alison El Ayadi, Nadia Diamond-Smith.

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
