## [Decision Letter · Decision Letter 0]

22 Aug 2024

PONE-D-24-13391The role of community-level men’s and women’s inequitable gender norms on women’s empowerment in India: A multilevel analysis using India’s National Family Health Survey–5PLOS ONE

Dear Dr. Gopalakrishnan,

Thank you for submitting your manuscript to PLOS ONE. After careful consideration, we feel that it has merit but does not fully meet PLOS ONE’s publication criteria as it currently stands. Therefore, we invite you to submit a revised version of the manuscript that addresses the points raised during the review process.

We look forward to receiving your revised manuscript.

Kind regards,

Pintu Paul

Academic Editor

PLOS ONE

Journal Requirements:

Reviewers' comments:

Reviewer's Responses to Questions

**Comments to the Author**

1. Is the manuscript technically sound, and do the data support the conclusions?

Reviewer #1: Partly

Reviewer #2: Yes

2. Has the statistical analysis been performed appropriately and rigorously? 

Reviewer #1: I Don't Know

Reviewer #2: Yes

3. Have the authors made all data underlying the findings in their manuscript fully available?

Reviewer #1: Yes

Reviewer #2: Yes

4. Is the manuscript presented in an intelligible fashion and written in standard English?

Reviewer #1: Yes

Reviewer #2: Yes

5. Review Comments to the Author

Reviewer #1: Thank you for this ms. This is well drafted. I have some provided some comments especially for the Introduction and Discussion sections. I have uploaded all comments separately, in a table format. All the best with the revisions.

Reviewer #2: Methods

The author mentions that there are 5 dimensions in measuring women empowerment. The author needs to explain in detail how these 5 dimensions form a single variable of women empowerment.

Authors should give the references how to measure each dimension of women empowerment.

Authors need to explain detail the data management of covariate variables

In methods, authors explain that education was measured as continues variables, however in the table 1, authors measured the education as categorical data

In page 8, authors stated “All these items were summed and dichotomized: women with greater than median score of correct answers were defined as empowered (coded as 1) and 0 otherwise”. From this sentence, I thought authors had 1 variable as women empowerment from composite of 5 dimension. However, when I checked in the result, authors did not do it. Additionally, in discussion, authors discuss about women empowerment, So I think it is better if authors also discuss form each dimension of women empowerment.

Results

Figure 1 is not clear. Authors should give the arrow direction to make causality between those factors

Table 3, Authors should explain detail how to measure the community-level men’s and women’s inequitable gender norms in the methods.

Table 4, please give title of column clearly

Table 5, authors did not include all variables in the model of multivariable analysis. Authors should explain detail how to select those variables in the statistical analysis in method section.

Table 5, authors categorized education based on education level in years. However, in table 1 authors presented education based on no education, primary, secondary, and higher education. Authors should be consistent when measuring the variable

Table 5, there is child ever born variable, however in the previous tables authors did not explain it or present it.

Discussion

In the first paragraph, authors should explain the general finding of study. In Table 5, authors analyzed the independent variables and each dimension of women empowerment. So, in the discussion section, authors need to sharpen the discussion based on the finding, compare with other studies, and explain possible mechanism.

Please add the policy implication based on your result in discussion section

6. PLOS authors have the option to publish the peer review history of their article (what does this mean?). If published, this will include your full peer review and any attached files.

Reviewer #1: **Yes: **Rakhi Ghoshal

Reviewer #2: No

---

## [Author Response · Author response to Decision Letter 0]

3 Oct 2024

To the reviewers and editors of PLOS ONE, 

We extend our sincere gratitude to the reviewers for their insightful comments and constructive feedback. The detailed suggestions have significantly contributed to improving the quality and clarity of our manuscript. We have carefully gone through the reviewer comments and responded to the reviewers’ comments in the last column of the table below. The formatting may not be okay here but I have uploaded this as an attachment. 

Reviewer #1: Comments

Section Text Response to reviewers’ comments

Background: Lower empowerment of women is a human rights issue with adverse public health implications. Though the concept of human rights in invoked in the abstract, in the main ms. at no point is this brought up. Human rights is too serious and important an issue and a concept to be invoked casually. Strongly suggest using only concepts that are used in the ms. to be used in the abstract. Changed the background section in abstract to below and addressed both the comments as mentioned by the reviewer.

 Would be very useful to have a few words to ‘locate’ the concept of gender norms in the Background, prior to invoking in in the Objective. 

Introduction 

Gender systems often tend to be patriarchal, ascribing greater value to masculinity rather than femininity Femininity is ascribed “greater value” on occasions such as child bearing or upholding the honour of the family. So, it is the context or the issue that determines if masculinity or femininity is ascribed greater value. I clarified this comment and elaborated more on how masculinity and femininity maybe context specific, but overall gender systems tend to be patriarchal. Also elaborated how masculinity is given greater value in most domains of social, economic, and political life and such patriarchal structures can persist even when femininity is valued in certain contexts.

nowhere else is this more relevant than in India. I would certainly say, it is way more relevant in Afghanistan or Palestine at the moment!

Well, my “reaction” aside, I suggest that we avoid such opinions that are not backed by solid evidence. If author(s) decide to use this line, they would then need to explain WHY is this most relevant for India. The “rational” provided at the moment convinces us why India should be studied, but not why this is most relevant for India among 192 countries. Apologize for the overstretching it and agree with the reviewer about such unsupported claims or those that are hard to prove. Instead, I rephrased the line to something more generic while acknowledging it’s a universal problem.

“Gender plays a crucial role in shaping various aspects of health and well-being across diverse societal contexts.” 

The most apparent evidence of this is India’s unevenly skewed sex ratios at birth (favoring boys) and the persistent favoritism towards boys displayed in care-seeking practices. Would be good to mention the sex ratio at birth, the latest data on that. while it might be there in the two references, it is better to embed it in the text itself. We have incorporated the latest available data on India's sex ratio at birth directly into the text. We have added the figure of 907 girls per 1,000 boys for 2018-2020 from the Sample Registration System (SRS), along with the appropriate citation 

In patriarchal societies such as India, we hypothesize that community-level gender norms are a byproduct of men’s beliefs about women’s roles and duties ––especially in the context of

cultural values, traditional family roles, gender values, and gender order. The basis of this hypothesis is not clear.

Evidence (from DHS in many countries, including India’s NFHS) shows that the % of women justifying wife beating is higher than % men justifying it. This, and a whole lot of other evidence shows that it is not just men who have harmful beliefs about women’s roles and duties; women themselves are convinced about their roles as home makers or the need to listen to the husband or take his permission for any major decision. So, I am not sure what is this hypothesis informed by. We have addressed these two comments by rephrasing this paragraph and moved all the hypotheses to one paragraph at the end of the introduction.

Since women also live in the same communities as men, we hypothesized that women’s attitudes towards 

gender equality might also add to the normative environment, which could influence women’s 

empowerment. We hypothesized that women residing in communities with higher inequitable men’s 

gender norms and inequitable women’s gender norms (modeled independently) will be more likely to 

have lower freedom of movement, lesser decision-making power, lower economic resources, and lower 

health empowerment. Would be good to “club” all the hypothesis at one place, at the end of the Introduction. 

Discussion

This underscores the significant influence of normative environment on women’s 

empowerment, irrespective of whether men or women uphold patriarchal norms and regressive attitudes. This sentence needs unpacking 

 I elaborated more on these results and elucidated the points better.

 Discussion section should speak more to the global literature - 

 Added more reviews and studies to highlight the adverse impact of inequitable gender norms.

This could plausibly be because even though women may be economically empowered 

with access to economic resources, inequitable gender norms could dictate the economic roles women 

should play Can the authors discuss a bit more on what they mean by “dictate the economic roles women should play”? what economic roles are women, even those with access to economic resources, expected to play? 

 Clarified what we meant this in better and clearer words.

In contexts where patriarchal systems are strong, such as India, 

analysis of community norms and their effect on various dimensions of women’s empowerment provides 

important evidence of the need to address women’s empowerment not only at the individual level but also 

at the community level. Can authors suggest some approaches for community level analysis of the various dimensions of women’s empowerment? We keep leaving all the difficult parts to “future research” and nobody gets around to that piece! So, if we can also suggest a few approaches or put forth some recommendations that are practical and doable, that would be a strong contribution to literature. 

 The reviewer is right that it’s crucial to provide concrete suggestions for community-level analysis rather than deferring to future research. I have expanded the paragraph significantly to include quantitative and qualitative approaches to measuring and analyzing dimensions of women’s empowerment at the community level .

….. highlighting the role of intergenerational transmission in the formation of gender attitudes. 

 Could the authors substantiate the argument about intergenerational transmission of gender attitudes a bit more? 

 We added more citations and substantiated the arguments around transmission of gender attitudes.

One study in India 

addressed social and gender norms to reduce women’s anemia rates…. Do we know what the intervention found? A line, reflecting on the evidence generated by this study, would be very helpful. 

 We added evidence from the study. 

we aimed to understand how the 

different sexes imbibe gender attitudes and impact women’s empowerment differently I am not convinced that the paper understands how the different sexes imbibe gender attitudes … the paper only studies the associations, not any in-depth analysis of the ‘how’ different genders imbibe gender attitudes … please edit this line. 

 We have edited this line. 

Consolidating women's 

empowerment into a single-scale measure presents several limitations, such as potential measurement 

errors and the risk of overlooking the multifaceted nature of the concept This is a “limitation” and should be part of the paragraph that talks of the Limitations. 

 We have corrected this.

Reviewer 2: Comments

Section/Text Response to reviewers

The author mentions that there are 5 dimensions in measuring women empowerment. The author needs to explain in detail how these 5 dimensions form a single variable of women empowerment.

 Apologies for the typo – it was four variables. Each of the measures have been mentioned in detail including construction on Page 8. 

Authors should give the references how to measure each dimension of women empowerment. Addressed this by citing previous papers that have used to measure different dimensions of women’s empowerment. 

Authors need to explain detail the data management of covariate variables. Our expanded explanation provides a more detailed account of your variable selection process and the data management of all covariate variables. We hope this satisfies the theoretical and empirical basis for our choices, as well as the statistical considerations involved.

In methods, authors explain that education was measured as continues variables, however in the table 1, authors measured the education as categorical data. Thank you for this observation. The reviewer is correct that there appears to be a discrepancy between our methods section and Table 1 regarding the measurement of education. To clarify:

1. In our statistical analyses, we indeed used education as a continuous variable, as stated in the methods section. This approach allows us to capture the full range of educational attainment and its nuanced effects on our outcomes of interest.

2. However, in Table 1, which presents descriptive statistics, we chose to display education as categorical data. This decision was made for several reasons: a) To provide readers with a clearer snapshot of the educational distribution in our sample. b) To allow for easier interpretation of the general educational landscape among our participants. c) To facilitate comparisons with other studies that often report education in categorical terms.

In page 8, authors stated “All these items were summed and dichotomized: women with greater than median score of correct answers were defined as empowered (coded as 1) and 0 otherwise”. From this sentence, I thought authors had 1 variable as women empowerment from composite of 5 dimension. However, when I checked in the result, authors did not do it. Additionally, in discussion, authors discuss about women empowerment, So I think it is better if authors also discuss form each dimension of women empowerment. The sentence on page 8 refers specifically to the health empowerment index, which is one of the four dimensions of women's empowerment we studied. We apologize for any confusion this may have caused. The reviewer is correct that we did not create a single composite variable for women’s empowerment from all four dimensions. Instead, we analyzed each dimension separately. Further, we have added some more depth to the discussion section by discussing each dimension of women's empowerment separately. Please note discussion section had to significantly rewritten to maintain flow of arguments.

Results 

Figure 1 is not clear. Authors should give the arrow direction to make causality between those factors. Thank you for your insightful comment on Figure 1. We appreciate your suggestion to add directional arrows to clarify the relationships between factors. However, after careful consideration, we have decided to maintain the current structure of Figure 1 without adding arrows for the following reasons:

First, Figure 1 is intended to be a conceptual framework that illustrates the multi-level nature of factors influencing women's empowerment. It is not designed to represent causal relationships directly. We have already included a Directed Acyclic Graph (Figure 2 in our manuscript) that explicitly represents the causal nature of relationships between key variables in our study. The DAG provides a more appropriate and detailed representation of the hypothesized causal pathways. Second, the relationships between factors at community, household, and individual levels are complex and often bidirectional. Adding arrows might oversimplify these nuanced interactions. 

To address your concern about clarity, we have added a note in the caption of Figure 1 explicitly stating that it is a conceptual framework and that causal relationships are represented in the DAG (Figure 2).

Table 3, Authors should explain detail how to measure the community-level men’s and women’s inequitable gender norms in the methods. We request the reviewer to please refer to pages 9 and 10 that explain the measurements of community-level men’s and women’s inequitable gender norms.

Table 4, please give title of column clearly. We appreciate your suggestion to improve the clarity of the column titles. We have revised the table to include more descriptive column headers. We have also added a row for each empowerment dimension to make the information more accessible.

Table 5, authors did not include all variables in the model of multivariable analysis. Authors should explain detail how to select those variables in the statistical analysis in method section. Regarding your comment on Table 5, we would like to respectfully point out that all variables included in our multivariable analysis are indeed present in the table. Perhaps there was some confusion or oversight in reviewing the table. We have double-checked Table 5 and can confirm that it includes all variables used in our multivariable analysis. The variables in Table 5 correspond directly to those described in our methods section, where we detailed our variable selection process. If there's any aspect of our variable selection or analysis that you feel needs more explanation, we would be happy to expand on it in the methods section.

Table 5, authors categorized education based on education level in years. However, in table 1 authors presented education based on no education, primary, secondary, and higher education. Authors should be consistent when measuring the variable. The reviewer is correct that there is a difference in how education is presented between these tables. This difference is intentional and serves different purposes in each context:

- In Table 1 (Descriptive Statistics): We presented education in categories (no education, primary, secondary, and higher education) to provide a clear, easily interpretable overview of the educational distribution in our sample. This categorical presentation allows readers to quickly grasp the general educational landscape among our participants.

- In Table 5 (Multivariable Analysis): We used education as a continuous variable (years of education) in our regression models. This approach allows us to capture the full range and nuance of educational attainment and its effects on our outcomes of interest. Using education as a continuous variable in regression analyses is a common practice in social science research as it preserves more information and can provide more precise estimates of education’s effects.

Table 5, there is child ever born variable, however in the previous tables authors did not explain it or present it. I revised the term ‘parity’ instead with number of children born. These terms are often used interchangeably in demographic and public health research, with parity being the more technical term referring to the number of times a woman has given birth to a fetus with a gestational age of 24 weeks or more. 

Discussion 

In the first paragraph, authors should explain the general finding of study. In Table 5, authors analyzed the independent variables and each dimension of women empowerment. So, in the discussion section, authors need to sharpen the discussion based on the finding, compare with other studies, and explain possible mechanism. We have rewritten most of the discussion section per reviewer’s comment. The first paragraph also summarizes the general findings of the study.

Please add the policy implication based on your result in discussion section. Policy discussion was written but now it has been revisited to make it impactful as suggested by reviewer.

---

## [Editor Report · Decision Letter 1]

8 Oct 2024

The role of community-level men’s and women’s inequitable gender norms on women’s empowerment in India: A multilevel analysis using India’s National Family Health Survey–5

PONE-D-24-13391R1

Dear Dr. Gopalakrishnan,

We’re pleased to inform you that your manuscript has been judged scientifically suitable for publication and will be formally accepted for publication once it meets all outstanding technical requirements.

Kind regards,

Pintu Paul

Academic Editor

PLOS ONE
---

## [Editor Report · Acceptance letter]

14 Oct 2024

PONE-D-24-13391R1 

PLOS ONE

Dear Dr. Gopalakrishnan, 

I'm pleased to inform you that your manuscript has been deemed suitable for publication in PLOS ONE. Congratulations! Your manuscript is now being handed over to our production team.

Kind regards, 

on behalf of

Dr. Pintu Paul 

Academic Editor

PLOS ONE